# Targeted Mop up and Robust Response Tools Can Achieve and Maintain Possum Freedom on the Mainland

**DOI:** 10.3390/ani12070921

**Published:** 2022-04-04

**Authors:** Briar Cook, Nick Mulgan

**Affiliations:** 1Tasman District Council, Richmond 7020, New Zealand; 2Zero Invasive Predators, Wellington 6012, New Zealand; nick@zip.org.nz

**Keywords:** brushtail possum, elimination, reinvasion, natural barriers, detection, response, New Zealand

## Abstract

**Simple Summary:**

To achieve the goal of Predator Free 2050, new pest control methods must be found and existing ones refined. We achieved the first >11,000-hectare possum elimination (all possums removed, and re-invaders subsequently mopped up) on mainland New Zealand at an unfenced site, largely protected from possum reinvasion by mountains and rivers. This was within a project to eliminate rats, possums and stoats. We spread a biodegradable poison called sodium fluoroacetate twice by helicopter and used existing ground-based methods (leghold traps, cage traps and a dog) paired with new technology (trail cameras and remote reporting of traps instead of manual checking) to maximise response efficiency. Possums were eliminated in 11 months from 11,642 hectares at a cost of NZD 1.6 million (1080 costs for all 3 species combined and ground-based costs for possums only). New possums entered the site at a rate of four per year and were detected and caught before a breeding population could establish. This proof of concept can now be repeated to confirm its success elsewhere, refine methods, and reduce costs. In the future, if possums are the sole target, we will likely spread 1080 once only and use effective mop-up tools to finish the job.

**Abstract:**

Unfenced sites on mainland New Zealand have long been considered impossible to defend from reinvasion by possums, and are thus unsuitable for eradication. In July 2019, we began eliminating possums from 11,642 ha (including approximately 8700 ha of suitable possum habitat) in South Westland, using alpine rivers and high alpine ranges to minimise reinvasion. Two aerial 1080 (sodium fluoroacetate) applications, each with two pre-feeds, were used. Here, we detail the effort to mop up existing possums and subsequent invaders in the 13 months following the aerial operation. Possums were detected and caught using a motion-activated camera network, traps equipped with automated reporting and a possum search dog. The last probable survivor was eliminated on 29 June 2020, 11 months after the initial removal operation. Subsequently, possums entered the site at a rate of 4 per year. These were detected and removed using the same methods. The initial elimination cost NZD 163.75/ha and ongoing detection and response NZD 15.70/ha annually. We compare costs with possum eradications on islands and ongoing suppression on the mainland.

## 1. Introduction

The introduced brushtail possum (*Trichosurus vulpecula*) is a major threat to both biodiversity and agriculture in New Zealand [1,2,3]. Having evolved in the absence of terrestrial mammals, New Zealand’s biodiversity is vulnerable to browse [4] and predation [5] by possums. As the primary vector of bovine tuberculosis (TB) in New Zealand, possums have the potential to severely impact the NZD 14.7 billion dollar per annum (all costs in this article are in 2021 New Zealand dollars and are exclusive of Goods and Services Tax (GST), unless otherwise stated) beef and dairy industries [6], while also depleting pasture and crops [7,8]. In 2015, NZD 79 million was spent on TB control alone [9]. In 2008, a report for the New Zealand Ministry for Agricultural and Fisheries estimated that invasive pests cost the primary sector NZD 2.13 billion annually [10]. 

Until recently, mainland elimination of possums over 10,000 ha or more had not been attempted, as there is a low chance of defending against reinvasion using existing tools. 

Possums have successfully been eradicated from several small islands and three large islands to date: Kapiti Island, Rangitoto-Motutapu Island and Whenua Hou/Codfish Island [11]. Possums on Kapiti Island (1967 ha) were eradicated in 1987, after eight years of intensive trapping, dog use and hunting [12]. Aerial sodium fluoroacetate (1080) was used to target animals along the western cliffs which were inaccessible on foot. This eradication cost approximately NZD 800,000 in 1987 [13], comparable to around NZD 1.8 million or NZD 864/ha in 2021. New Zealand inflation is calculated using the Consumer Price Index [14]. The multi-species eradication that included possums on Rangitoto-Mototapu Island (2333 and 1510 ha respectively) began in 1990. It took four years to remove possums from Mototapu and six years to remove them from Rangitoto [15]. Aerial 1080, cyanide (both in bait stations and conventionally placed on trees and the ground), leghold traps and dogs were used. The Whenua Hou Codfish Island possum eradication (1396 ha) began in 1984 and took three years [16]. Toxins, trapping and dogs were used [17].

Areas of the mainland where predators have been removed are susceptible to reinvasion. One tool developed to combat this is the predator fence, which has made it possible to achieve multi-species eradications on the New Zealand mainland. The largest fenced site is Sanctuary Mountain Maungatautari [18]. In 2006–2007, two applications of aerial brodifacoum were applied to 3400 ha inside a 47 km long predator ring fence. The initial knockdown was followed by monitoring and targeted mop up of survivors. Possums, European hedgehogs (*Erinaceus europaeus*), feral pigs (*Sus scrofa*), ship rats (*Rattus rattus*) and goats (*Capra hircus*) were successfully eradicated [18]. 

Despite these successes, there are drawbacks to predator-fenced sanctuaries. A predator fence is not impermeable—weather events and tree falls can create breaches. A trial at Sanctuary Mountain Maungatautari revealed that artificial fence breaches were found by predators within 24 hours [18]. Fences must be checked regularly to rapidly identify such events and prevent incursions. 

Predator fence construction costs NZD 327– NZD 506/linear metre [19]. The Sanctuary Mountain Maungatautari predator fence cost an average of NZD 402/linear metre or NZD 5615/ha [20], excluding ongoing maintenance. Maintenance is approximately 4% of the initial cost annually [21]. The level of sustained management involved in such sanctuaries is not generally considered scalable above a few thousand hectares [22]. 

The second-generation anticoagulant brodifacoum has been successfully used for aerial rodent eradications on off-shore islands [23] and for control of possums and rats using bait stations on the mainland [24]. Aerial application of brodifacoum on the New Zealand mainland is restricted to predator-fenced sites [25] due to concerns about non-target impacts, toxin persistence in mammals and human consumption of contaminated meat [26,27]. 

Repeated, ongoing application of aerial 1080 is the most common approach used at large (>10,000 ha) scale on the New Zealand mainland to suppress pest populations to low numbers [24,28]. While this cyclical approach benefits many native species [29], the costs are ongoing and results are limited by the cyclical recolonisation and breeding of pest species [30].

While the complete eradication of invasive pests is generally accepted to achieve the greatest biodiversity gains and relief from disease transmission, factors such as terrain, labour costs, ongoing surveillance, biosecurity response and reinvasion certainty make eradication at scale on mainland New Zealand extremely difficult. New techniques are required [21,22,31] to meet the Predator Free New Zealand goals [32] of eradicating rats, stoats (*Mustela erminea*) and possums by the year 2050. 

In the large-scale mainland setting, it is helpful to consider the epidemiological concept of ‘elimination’ [33], as opposed to eradication. Elimination recognises that a small number of animals may exist within a predator free site until they are detected and removed. As long as an increase in numbers to a self-sustaining population is prevented, and the population is repeatedly reduced to zero, then the elimination state persists. Clearly this requires both sufficiently quick and reliable detection and targeted removal. Eradication can be defined as the removal of every individual of a species or the reduction of their population density below sustainable levels, with reinvasion prevented. The distinction between the two definitions is subtle, but important.

Possum reinvasion can be reduced by selecting sites that utilise natural barriers to movement, such as alpine ranges and cold, fast rivers [30,34,35]. Here, we present a case study illustrating this approach to possum elimination at the >11,000 ha scale in the Perth River Valley, South Westland, New Zealand, focusing on detection and removal. We estimate the costs of achieving and maintaining possum elimination at this site and compare with previous island eradications and the repeated suppression status quo.

## 2. Materials and Methods

### 2.1. Study Site

The Perth River valley study site (43.2616° S, 170.3590° E) (Figure 1) overlaps the 46,587 ha Adams Wilderness Area, 16 km southeast of Whataroa township, South Westland. The 11,642 ha site is bordered by the Southern Alps to the northeast, east and south, the Perth and Bettison Rivers to the southwest and the Barlow River to the west and north. This site was deliberately chosen for possum elimination due to the protection afforded by the cold, fast flowing rivers, which make up approximately 19 km of the site boundary. The remainder is protected by permanent snow and ice, with the exception of the headwaters of one river and an alpine pass (discussed below). 

Altitude in the study site ranges from 200 m to 2200 m above sea level (asl). Possums have been seen above 2000 m asl at comparable sites [36]. 

The predominant forest type is southern rātā (*Metrosideros umbellata)* and kāmahi (*Weinmannia racemosa*) forest, with additional broadleaf podocarp forest containing scattered pockets of rimu (*Dacrydium cupressinum*). Kōtukutuku/tree fuchsia (*Fuchsia excorticata*), a species consumed by possums in South Westland [37], is also present. Above 1200 m asl, alpine scrub and tussock dominate, eventually giving way to scree and sheer rock faces. Valued fauna species include pīwauwau/rock wren (*Xenicus gilviventris*) and kea (*Nestor notabilis*) which are both nationally endangered, whio/blue duck (*Hymenolaimus malacorhynchos*) (nationally vulnerable) and kārearea/New Zealand falcon (*Falco novaeseelandiae*) (recovering). 

In 2019, Zero Invasive Predators (ZIP), a not-for-profit research and development entity, launched an elimination project in the Perth River Valley study site. Prior research, including the effectiveness of dual 1080 operations [38] and bait switching techniques [39] to achieve local elimination of possums and rats, helped inform the methods used in this project.

The initial removal included two aerial 1080 operations and was completed in July 2019. Following the operations, possum detections in the site reduced by more than 99%. Details of the aerial predator elimination operation and results up to seven weeks after the operation are described in [40]. 

### 2.2. Detection 

A network of 142 Browning Dark Ops™ (Prometheus Group, Birmingham, AL, USA) trail cameras were installed at a spacing of approximately 700 m × 500 m, for a density of approximately 1 per 35 ha (Figure 1) by March 2019. This density was designed to be economically viable at large scale, while still able to detect ship rat breeding populations. It was also expected to be sufficient to detect an individual possum exhibiting the increased ranging behaviour associated with those isolated from conspecifics [41,42]. The cameras were set to take three photos per trigger event (3-shot rapid fire, with a 5 s delay between triggers) to increase the probability that at least one of the images captured the target animal. The location, date and time were recorded for all images. Cameras were equipped with 16 G Sandisk™ SD cards, and six AA rechargeable batteries, both of which were replaced every 6−8 weeks. All footage was reviewed within a few days of collection and images of target species were identified and recorded.

In January 2020, an additional five lured trail cameras were installed along the true left of the river boundary at the Barlow headwaters, in anticipation of reinvasion pressure from the adjacent Poerua Valley. 

Each camera was mounted on a tree or waratah at 1.5–2 m distance from a ZIP MotoLure™ (ZIP, Wellington, New Zeeland), an automated lure dispenser designed to exude 0.15 mL of fresh lure into the environment every night [43]. This device rewards repeat visits, thereby increasing detectability. Best Foods Mayonnaise™, found to be attractive to rats, possums and stoats (ZIP, unpublished data), was used in the lure dispensers. 

A possum dog was used to help identify the most recent location of a target animal for deployment of response tools following camera detections. 

### 2.3. Supplementary Aerial Operation

A supplementary aerial 1080 operation aimed at removing an emerging ship rat population was carried out over 863 ha (7.4% of the block) on 5 March 2020 (Figure 1). The operation consisted of three prefeed applications at 1 kg/ha and a single toxin application (0.15% 1080) at 2 kg/ha. All used 6 g RS5 bait lured with double cinnamon (Orillion, Wanganui, New Zeeland). 

### 2.4. Detectability of “Roaming” and “Settled” Behaviour—Upper Perth Survivors

The relative isolation of the upper Perth Valley provided an opportunity to measure the effectiveness of trail cameras paired with a lure dispenser for survivor detection in an effectively closed system. Two possums appeared to range over tens of hectares when unaware of each other, and then to ‘settle’ in a smaller area together once they found each other. By assuming that detectability times area is constant [44], we estimate the size of this settled area. 

The sample detection probability per night is
(1)$$p=n_{detect}/\left(N\times n_{nights}\times \;n_{cameras}\right)$$
where *n_detect_* = number of trap nights which included a detection, *N* = estimated number of possums, *n_nights_* = number of nights over which detections occurred, *n_cameras_* = total number of cameras in the area. 

We used Voronoi polygons [45] centred on the cameras and the Upper Perth River to approximate the home range of surviving possums in the upper Perth Valley, and calculated detectability before and after aggregation.

To compare with other results and theory, we note that spatially explicit capture-recapture theory [46,47] uses nightly probability of detection:(2)p=g0e^ −r^2/2σ^2 
where *g*_0_ is the probability of detection if the detecting device is at the centre of the animal activity area, *r* is the distance from the device to the centre of the activity area and *σ* is the radius of the activity area. This gives a total detection x area of 2 *πσ*^2^
*g*_0_. [46] give *g*_0_ = 0.17, and *σ* = 63 m for 2 *πσ*^2^
*g*_0_ = 0.42 ha/night for possums caught in leghold traps. 

### 2.5. Frequency of Invaders

To calculate a confidence interval for the mean annual invasion rate, we suppose temporarily that the arrivals were uncorrelated. In that case the Poisson distribution is appropriate. We use the mid-P method to calculate a 95% two-sided confidence interval for the mean annual reinvasion rate.

## 3. Results

### 3.1. Detections

In the six months following the two initial 1080 operations (July to December 2019), possum detections were restricted to two areas, the Lower Barlow Area and the Upper Perth Valley (Figure 2 and Figure 3). 

In 2019, the Lower Barlow detections were limited to two detection lines, approximately 700 m apart. Early in 2020, a trail of detections spread to the Perth-Barlow confluence, with no noticeable reduction in the frequency of detections on the original two lines. There were 62 distinct camera–day detections between the second removal operation and the supplementary toxin operation (July 2019 to March 2020) in the Lower Barlow and Confluence areas. At the time of the supplementary operation, the detection trail was spreading up the true right side of the Perth River, towards the location where a decomposed possum body was found on 19 May 2020 (Figure 2 and Figure 4). 

In the Upper Perth Valley, not subject to the supplementary aerial operation, there were sporadic detections in 2019 (8 camera–days) and 2020 (11 camera–days). Two possums were caught in cage traps on 11 June and 29 June 2020 (Figure 4). No possums were detected in the Upper Perth Valley for the remainder of 2020. 

Between 7 March and 31 August 2020, three probable invaders were detected and captured in cage traps in the Barlow Headwaters (Figure 4). From 31 August 2020 to 1 July 2021, a further five possums were caught in the Barlow headwaters (not shown).

Data regarding camera servicing, device trap nights and targeted response effort are summarised in Table 1. The data are divided into probable survivor mop up and probable invader response. It is not possible to be completely certain whether a particular possum is a survivor or an invader, though timing, location and behaviour help infer these classifications.

### 3.2. The Cost of Possum Elimination

The approximate cost of the possum elimination including the aerial 1080 operations and mop up of existing and subsequent possums totalled approximately NZD 1.6 million, or NZD 163.75/ha (excluding GST). The breakdown of this cost is summarised in Table 2. Note that this work was carried out within a wider Perth Valley project for rats and stoats as well as possums, so attribution of costs, especially helicopters for staff transport, is only approximate.

### 3.3. Detectability and “Roaming” vs. “Settled” Behaviour

Over 11 months following Phase 2 of the 1080 to Zero operation (23 July 2019–29 June 2020), there were 19 distinct camera–night detections on five out of eight cameras adjacent to the river in the Upper Perth Valley (Figure 2 and Figure 5). During the same time period, there were no detections on the 15 other cameras covering four kilometres to the only natural exit to the valley, so it is very likely that this valley represented a closed system. Two possums, an adult female and a sub-adult male, were captured in the Upper Perth Valley in June 2020, with no further detections. 

In the first nine months of this period, between 23 July 2019 and 14 April 2020, ten detections were spread across the five cameras (Figure 5a). It was not possible to tell the two animals apart in all images, so we cannot rule out the possibility that both animals were ranging over the entire area between the detections. Detectability for this time period is calculated as 0.0023/TN with a 95% CI of (0.11, 0.42) for both animals in the combined area. A second possibility is that the activity area for each possum was separated by Tarn Creek (Figure 5). Detectability would then become 0.0046 with 95% small sample CI of (0.23, 0.84). Detectability is doubled as each possum is assumed to not have access to half of the cameras. The approximate range of the two animals was calculated as 93 ha, or approximately half of that each if occupying separate areas. The product of detectability and area will be unchanged [44]. The uncertainty in the area is large—we estimate ±50%. 

From 15 April 2020 up to and including captures, there were a further nine detections, all at a single camera (Figure 5b). Based on this and the two capture locations, we can assume that the possums were not separated by Tarn Creek. Over this time period, detectability increased to 0.026/TN, 95% C.I. (0.07, 0.44).

## 4. Discussion

Survivor Detectability—Upper Perth Valley

The very low frequency of detections in the Upper Perth Valley (relative to the Lower Barlow), the detection locations and the lack of detections following the two captures led us to believe there were exactly two animals present. 

We hypothesise that the reduction in the spread of detections from 15 April–29 June 2020 occurred when the two animals became aware of each other. It also corresponds with a period of lower river flows which may have made the intervening Tarn Creek (Figure 5) temporarily easier to cross. 

The nightly probability of detection was calculated as (0.07, 0.44) ha/night, just consistent with the theory of 0.42 ha/night for possums [46,47], i.e., the two possums were relatively difficult to detect.

Lower Barlow Area

Because there were no detections in the Lower Barlow Area after the supplementary aerial operation (March 2020), it seems probable that all possums in this area were killed in the supplementary operation and that there were no invaders across the lower Barlow and Perth rivers.

### 4.1. Frequency of Invaders

There were no possum detections in the Barlow headwaters between the first aerial removal operation on 13 April 2019 and 6 March 2020 (Figure 3). This seems an unusually long period for a survivor to be present and undetected, when survivors in other areas were discovered within a few months of the operation. Hence, all three possums caught in the Barlow headwaters in 2020 are believed to have been invaders either over the pass (1520 m asl) from the Poerua catchment, or from the true right of the upper Barlow River Valley, across the river. The proximity to the Poerua catchment and the extremely low level of the headwaters at certain times of the year (at the end of the thaw and during winter when water is tied up in ice) supports this hypothesis. Together with the five possums caught in the following year, this equates to eight possums spread over two years. There are multiple methods appropriate for calculating a Poisson confidence interval, for the mean annual invasion rate, which give similar values from a sample of eight [48]. The mid-P method gives (3.72, 15.19) or (1.9, 7.6) per annum for a 95% two-sided confidence interval. In actuality, it is probable that the arrivals will be correlated, likely positively, increasing the variance, so that a somewhat wider range would be appropriate. 

The constant reinvasion pressure highlights the difference between island eradications and mainland elimination, and the importance of detection and response plans on the mainland. While the reinvasion level here is manageable, it may be greater at other sites.

As immigration by possums following population control has consistently been shown to be by young animals, and mostly males [30], we expected a priori that the invading possum demographic would largely be made up of sub-adult males dispersing into new territory. However, of the eight invaders captured to 1 July 2021, only one was a sub-adult male, five were adult males and two were adult females. It was also notable that possums appeared to reliably seek out the same small patch of bush on a ridgeline on the true left of the Barlow headwaters. It is possible that this patch is seen as the best habitat in the area by possums. It is also likely that calls, urine trails or other sign directed the animals to conspecifics. It could be informative to catch possums on the true right of the Barlow headwaters and in the upper Poerua Valley and fit them with GPS or VHF transmitters in order to assess their movement behaviour in detail. This may reveal further insight into when and how they invade the Perth Valley site, and why they appear to favour a particular area.

An aerial thermal hunting survey (Stay Put Ltd., Aerial Thermal Hunting Services, Whakatane, New Zealand) for invaders in October 2020 detected a probable possum invader in the same patch of forest mentioned above. No other possums were detected in approximately two hours of survey covering both sides of the Barlow River headwaters. Thermal cameras may be a useful tool for identifying and removing the last few individuals.

### 4.2. Detection and Response Tools

Lure dispensers paired with trail cameras made for efficient possum detection. The ability of the dispenser to keep a target animal returning to a place where it can reliably be detected [49] without ill effects is promising. The use of motion sensing cameras paired with these devices in the site cut out the labour traditionally required to manage intensive trapping programmes, lay monitoring lines and manually search large areas for survivors and invaders. At a large scale, the ability to avoid this intensive work is critical—there is too much ground to cover otherwise. 

The next development is to pair the lure dispenser with a long-life, artificially intelligent (AI) camera that carries out on-board analysis of each recorded image, then notifies field staff within 48 h via a remote reporting system of the presence of a target animal. If successful, this will significantly reduce the labour involved in manually servicing trail cameras, and improve the field response time. ZIP is currently trialling such a system in the Whataroa Valley, adjacent to the Perth Valley study site.

### 4.3. Lure Dispensers and Cage Trap Efficacy

Because lure dispensers were used in the detection network, it made sense to deploy the same system on the live capture cage traps, as detected animals had already interacted with the lure. Live capture traps must be able to be accessed within 12 hours of sunrise on any day they are set for animal ethics reasons [50]. Because of the remoteness of the site, field staff worked an eight-day stint in the site with four days off. During the four-day breaks when cage traps could not be set, they were instead wired open and left with lure dispensing into the back of the cage, at a high rate of 1 mL/day where possible (compared with 0.15 mL/day for dispensers paired with cameras). On return to the site, it was not unusual to catch a possum on the first night of trapping. It seems that this pre-feeding reward increased the efficacy of the cage traps, reducing the number of trap nights required to target each possum.

### 4.4. Automated Reporting

Automated reporting of leghold and cage traps cut down labour requirements and transport costs. In a site requiring helicopter access, flying to check traps every day was expensive. The ability to remotely monitor the traps using satellites, low-powered radio signal and a web server meant that set traps were only serviced when needed.

The ability to carry out a targeted response to detections, effectively pre-feed traps and automate reporting of captures enabled the timely and efficient elimination of surviving and invading possums from the Perth Valley study site. 

### 4.5. Supplementary Aerial Treatment

The coincidental supplementary aerial 1080 treatment (targeting rats) helped remove possums faster. Exactly why the remaining possums succumbed to this operation (but not the initial operations) is unclear. The supplementary operation occurred six months after the removal operations, and was pre-fed three times instead of two. The possums were presumed to encounter and consume the bait in the absence of high densities of rats and other possums. Any combination of these factors may have improved the success of the operation enough to eliminate the Lower Barlow survivors. The evidence from response actions in the Barlow Headwaters and Upper Perth Valley leads us to believe that the possums killed by the supplementary aerial treatment in the Lower Barlow and Confluence areas would eventually have been caught using ground-based methods. Further studies are needed to confirm this, and until then, the duration and cost of ground-based mop-up remain difficult to estimate. 

### 4.6. Operational Cost of Possum Elimination

The Kapiti Island eradication effort cost approximately NZD 800,000 in 1987 [13] comparable to NZD 1.7 million in 2021 [51]. This cleared 1967 ha for NZD 864/ha (in 2021 currency), over eight years. In the Perth site, for NZD 1.6 million or NZD 163.75/ha possums were removed from 11,642 ha over one year. Of course, no two sites are equivalent, but this difference does suggest that the newer methods are quicker and more cost-effective. The progress is aided by the fact that much has improved since 1987, including best practice methods for vertebrate pest control. The obvious disadvantage is that subsequent invasions will eat into any cost efficiencies. There is potential for costs to blow out where survivorship and invasion rates are unmanageable, or the elimination takes longer than initially planned (such as the rat elimination on the Miramar Peninsula). This potential can be minimised with careful site selection—including in-depth site research to confirm suitability before any control is undertaken; precise operational timing, constant reassessment of progress and the ability to rapidly adjust to unexpected setbacks. This is a new area of work, so learning from other projects is vitally important. 

After more than two years at a rate of four possums per year in the Perth Valley, and a targeted elimination cost of NZD 15/ha, maintenance costs at this site are very modest. The intention is that elimination at bordering sites as part of the Predator Free South Westland project will reduce possum invasion pressure on the Perth Valley site, and the lack of existing possums in the site will in turn reduce pressure on successive sites in a rolling elimination front. The cost per hectare will shrink as the proportion of elimination front to the core protected area diminishes.

### 4.7. Cost Comparison with Suppression Operations

Aerial 1080 operations to suppress possums (and rats and stoats) cost between NZD 12–16 per hectare [52]. When compared with the present project, this seems very inexpensive. However, suppression operations must be repeated every few years and include the inevitable bounce back of possum populations in the years between operations. The repeated application of aerial 1080 with the same lure and mask, while effective initially, can induce bait shyness in possums that receive a sublethal dose [53,54,55], reducing the efficacy of suppression operations over time. 

### 4.8. Research and Development Cost Component 

The methods used for the Perth Valley predator elimination were ZIP’s best estimate of what would successfully achieve the elimination objective at the time. This was the first >11,000 ha predator elimination attempt of its kind, so there were considerable research and development costs built into the operation. Some of the tools used were recently developed (e.g., the lure dispenser) or developed through the elimination process when it became apparent there was an urgent need (e.g., the automated reporting system adaptation for use with cage traps). Other tools were initially planned for use then removed completely as a result of trials in the site. Leghold traps were not used because a trial investigating kea interactions (with disabled traps) indicated a greater than expected capture risk to this species [56].

The aerial removal operation targeting rats, stoats and possums consisted of two phases, each made up of two pre-feed operations and a toxin operation. Carrying out six aerial operations instead of the typical two (one pre-feed, one toxin) understandably increases the cost. The other major difference in the Perth Valley aerial elimination operation was the 50% overlap between baiting swathes flown by each helicopter. This ensured there were no gaps in the operation, but also increased the flying time by 50% [40] (commute time unchanged, sowing time doubled).

### 4.9. Cost Refinements

Now that the Perth Valley elimination operation and subsequent mop-up has shown proof of concept, the next step is to repeat a similar operation and continue to refine the methodology for South Westland, with the goal of further increasing effectiveness while driving the costs down. Planned refinements include trialling a single toxin operation at high elevation, the AI cameras discussed above, lower device densities and toxic lure dispensers to swap in when predators are detected.

ZIP aims to reduce initial removal costs for other South-Westland sites, (for all three target species) to NZD 150/ha, detection costs to NZD 100/ha/annum and ongoing maintenance costs to less than NZD 10/ha/annum. 

The method will then need to be adapted to warm, humid forest sites, where abundant food sources maintain high densities of pest mammals, such as those found in the North Island of New Zealand. These may require higher bait densities or a staged approach. There will be sites where non-targets present challenges to the described methods used, and sites that are inhabited. Each unique situation will require adaptation, including possible further research and development work. 

## 5. Conclusions

Possums were eliminated from an 11,642 ha site in mainland New Zealand for the first time. The cost was approximately NZD 160/ha for initial elimination and NZD 15/ha annual maintenance.

In two years, reinvasion has been consistently geographically isolated to the headwaters of the upper Barlow River and the pass to the Poerua catchment (1520 m asl). The two possible entry routes are indistinguishable at present. The majority of the river boundary appears to be wide, fast and cold enough to deter possums that encounter it.

The success of the possum elimination is a boost for the Predator Free South Westland project, and the wider Predator Free 2050 goal. It has demonstrated that, at this site at least, it is possible to eliminate possums on the mainland. The site was defended from population re-establishment without the use of repeated large scale 1080 operations, and at reasonable cost. The site has now had over two years of ecosystem recovery, providing rich habitat for the native fauna and flora that live there. The successful elimination of possums has provided a springboard to refine the methodology for rats and stoats.

## Figures and Tables

**Figure 1 animals-12-00921-f001:**
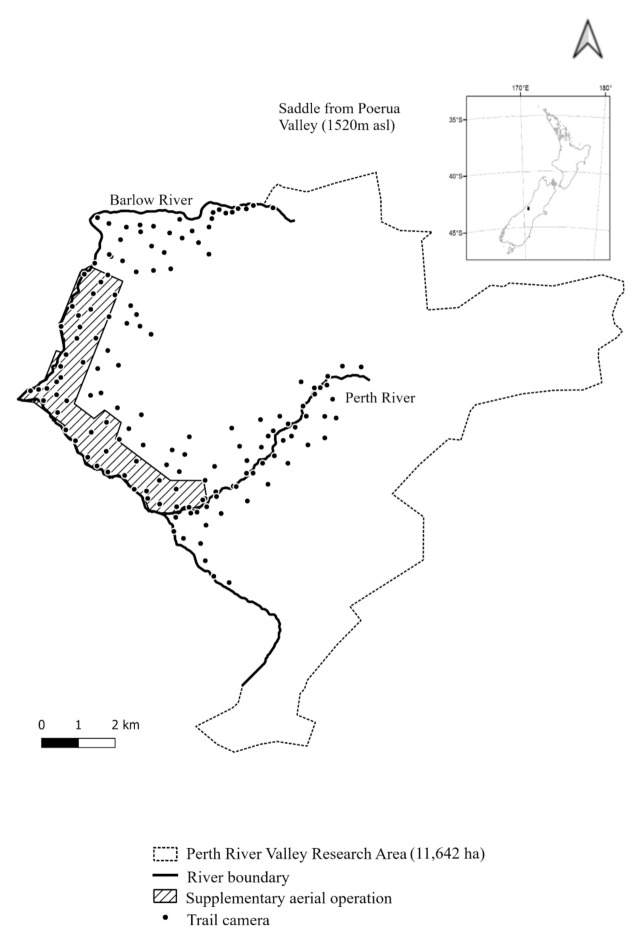
Study site showing trail camera locations, protective river boundaries and the supplementary aerial treatment coverage.

**Figure 2 animals-12-00921-f002:**
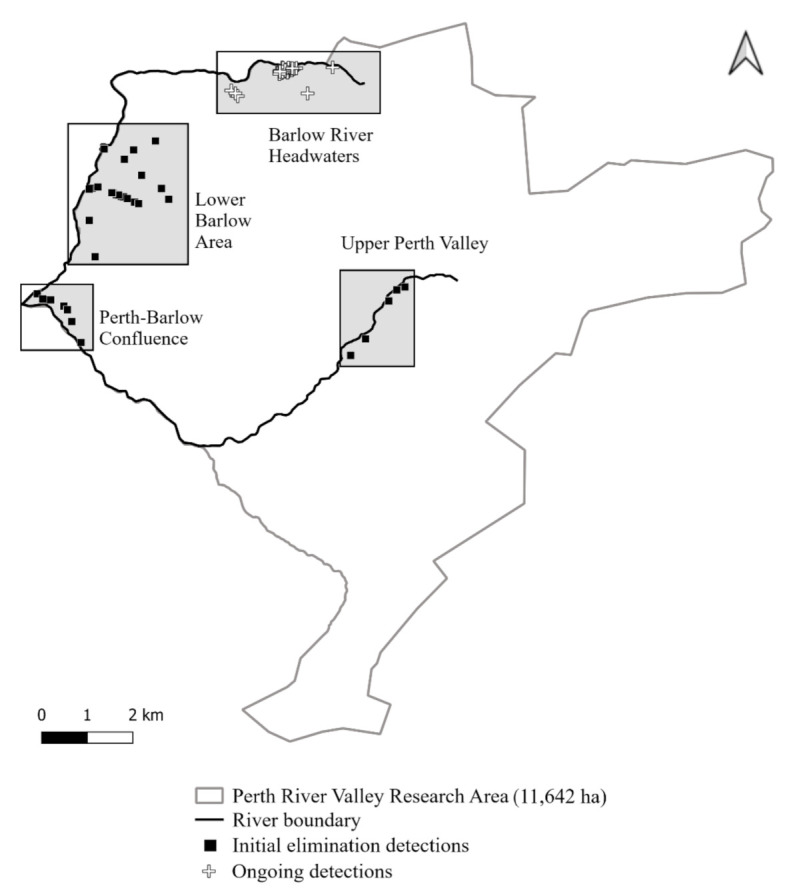
Possum detections in the Perth Valley from 25 August 2019 (first detection following site wide aerial 1080) to 31 August 2020. Shaded squares indicate the discrete areas of the site discussed in the text.

**Figure 3 animals-12-00921-f003:**
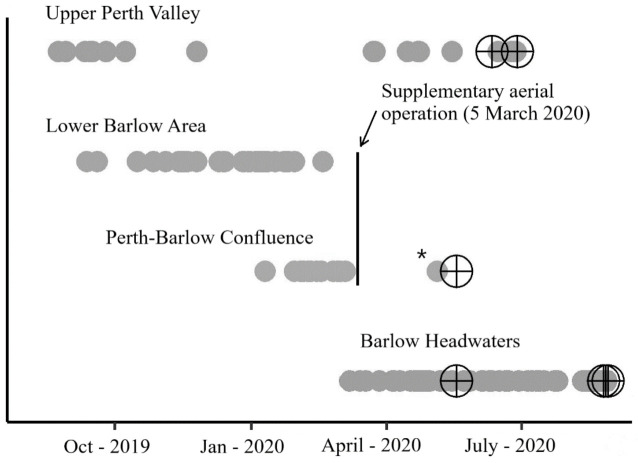
Possum detections by area over time. Crossed circles represent kills. * The Perth-Barlow confluence possum detection represented on the figure after the supplementary aerial operation was a possum scat, and the body found near the same location is believed to have died from that toxin operation.

**Figure 4 animals-12-00921-f004:**
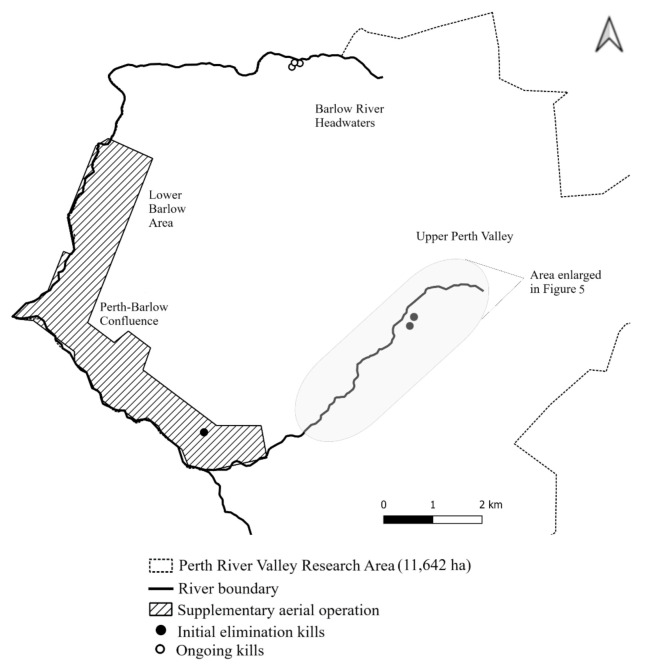
Kill locations of possums in the Perth Valley from 5 May (first body found) to 31 August 2020.

**Figure 5 animals-12-00921-f005:**
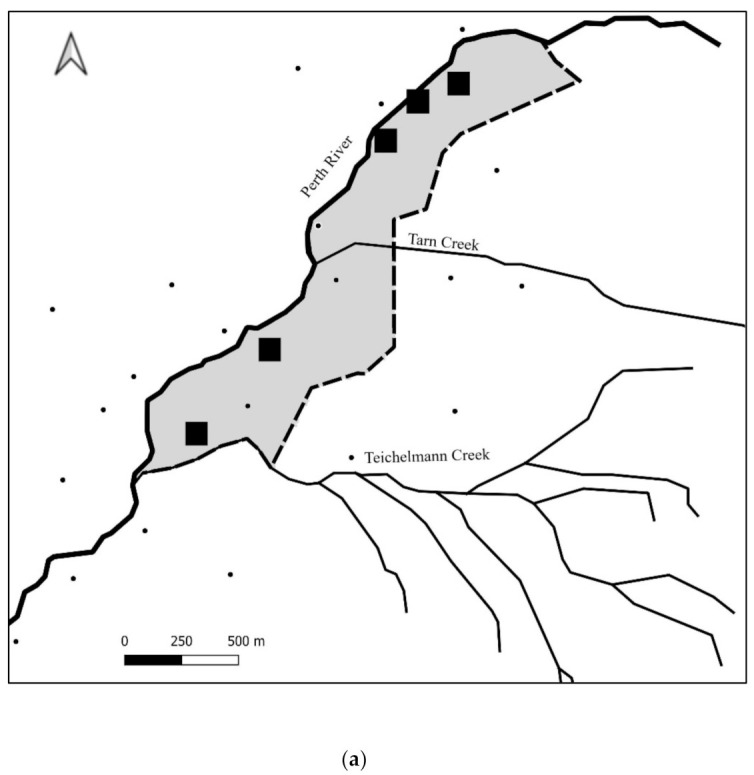
Detection, estimated home ranges and kill location for survivors in the Upper Perth Valley (area enlarged from Figure 4): (**a**) Detection locations 23 July 2019–14 April 2020; (**b**) detection and capture locations 15 April–29 June 2020. The number of detection locations differs from the number of detections stated during these periods, as camera sites were visited multiple times.

**Table 1 animals-12-00921-t001:** Trap nights (TN) and days of effort using detection and response tools for (assumed) surviving and invading possums in the Perth Valley.

Tool	Probable Survivor Mop Up	Probable Invader Response *	Total
	23 July 2019–29 June 2020	8 March 2020–31 August 2020	
Lured camera network (TN)	48,002	9650	57,652
Extra cameras × 5 (TN)	0	1145	1145
Remote reporting legholds (TN)	170	0	170
Remote reporting cage traps (TN)	192	60	252
Total (TN)	48,364	10,855	59,219
Possum dog (days)	48	19	67
Targeted person (days)	55	25	80
Camera servicing (days)	164	84	248
Total (days)	267	128	395

* Four standard cameras at the Upper Barlow were detecting invaders while the rest were still detecting survivors. This is accounted for from the first invader detection date, 7 March 2020. The remainder of cameras switched to invader detection on 29 June, the date the final survivor was caught.

**Table 2 animals-12-00921-t002:** Approximate costs of the Perth Valley possum elimination and ongoing response to 31 August 2020. Per ha costs are based on the 8659 ha treated in the first 1080 operation.

Initial Elimination Costs	2020 NZD Excl. GST
Aerial 1080 and kea mitigation	NZD 1148,435.54
Detection network and servicing	NZD 170,731.89
Supplementary treatment	NZD 54,004.04
Targeted ground response	NZD 44,782.61
Total	NZD 1417,954.08
Total per ha	NZD 163.75
Ongoing Costs	2020 NZD excl. GST
8 March to 31 August 2020	NZD 65,554.78
Equivalent annual cost	NZD 135,951.68
Annual cost per ha	NZD 15.70

## Data Availability

The data presented in this study are available on request from the corresponding author. The data are not publicly available yet.

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
