# Peer review of "Targeted Mop up and Robust Response Tools Can Achieve and Maintain Possum Freedom on the Mainland"

_animals, 2022, doi:10.3390/ani12070921_

Round 1
Reviewer 1 Report
Congratulations to the authors for a well written, interesting and useful paper.
My main comments are below; minor comments/corrections can be found in the document attached.
Definitions. Eradication, elimination, removal, are used without defining each term. Elimination is defined at the end of the introduction, but not the other terms. This could be misleading. For example, you write ‘New techniques are required [21, 22] to meet the Predator Free New Zealand goals [31] of removing rats, stoats and possums.’ The goal is in fact eradication, so clear definitions are needed.
Section 4.5. Survivors of the first 1080 operation were killed during the supplementary treatment. Why do you think they died in the second op but not in the first one? As you know, there is much debate on whether some animals either avoid the bait or simply are not able find bait. Your case may suggest it was the latter.
Section 4.6. While I agree with the arguments, there are 2 important points that warrant brief mention in the text. First, there is no doubt the Kapiti eradication would be cheaper and quicker if done today. Since 1987, the year of the eradication, much has improved, including best practice. Second, it’s important to clarify that mop up costs can quickly increase if the rates of survivorship/invasion are higher than in your case. Actually, the case of Taranaki, where progress is being much slower, deserves a comment.
Section 4.9. Again, this assumes that only reduction costs are expected. Are there any factors that may increase the costs? Any site in a warmer region (e.g. North Island) may need higher bait densities as most likely there’ll be more bait consumers (possums, rodents, invertebrates).

Reviewer 2 Report
The authors have elaborately presented the application of a method for eliminating possums from not fenced mainland areas. An informative and useful discussion of the benefits and possible improvement of the method that would be useful elsewhere.
Corrections during proofing
Line 12: Explain what is 1080 in the lay summary. The readers should not have to search in the manuscript for an explanation.
Line 22: Explain ZIP. This is the first time mentioned.
Reviewer 3 Report
Manuscript ID: animals-1636468
Title: Targeted mop up and robust response tools can achieve and maintain possum freedom on the mainland
By Authors: Briar Cook *, Nick Mulgan
Review
General remarks
From the Line 75 it could be implied, that two currencies, USD and NZ$ are used in the manuscript, this causes some troubles to compare, so please stick to one of these.
As Animals do not charge for colour figures, colours should be introduced for better figire readability.
CI calculation not presented in the Methods
To give international readers broader perspective, other possum eliminations with 1080 should be mentioned, such as:
Nugent et al.: Dual 1080 possum and rat control New Zealand Journal of Ecology (2019) 43(2): 3373 © 2019 New Zealand Ecological Society
Nichols et al.: Predator removal at large scale New Zealand Journal of Ecology (2021) 45(1): 3428 © 2021 New Zealand Ecological Society.
Nugent et al.: Bait switching for possum control New Zealand Journal of Ecology (2020) 44(1): 3403 © 2020 New Zealand Ecological Society.
and possibly others, about which readers could be not aware.
Also, for the international reader, more information on Predator Free 2050 should be presented, e.g.,
Ross, J. G., Ryan, G., Jansen, M., & Sjoberg, T. (2020). Predator-free New Zealand 2050: Fantasy or Reality?
Simple summary
Not all readers know that 1080 is the common name for a biodegradable poison called sodium fluoroacetate. Please use full name also in Simple summary
Abstract
Do not use ZIP, as unexplained abbreviation here
Keywords
Add New Zealand
Introduction
Line 37 and further: correct referring is [1–3]
Line 41 and further: correct referring is [7,8]
Line 43: “(2008) annually” is not correct, as “annually” implies current period also. Please rewrite
Line 65 and further: use long dash for range, 2006–2007
Line 68: please use correct common names for E. europaeus and S. scrofa
Line 95: please add scientific name for stoat at the first use
Results
Figure 5 could be merged with figure 4, putting these parts as insets into Fig. 4
Line 252: figure 5a did not show 10 detections, add legend or explain in the caption
Line 263: figure 5b did not show 9 detections, add legend or explain in the caption
Discussion
Lines 283–290 belong to Methods, thus should be transferred
Line 326: (David Paine) is unclear – is this personal communication?
Back matter
Please add statement on “Any role of the funders in the design of the study; in the collection, analyses or interpretation of data; in the writing of the manuscript, or in the decision to publish the results must be declared in this section. If there is no role, please state “The funders had no role in the design of the study; in the collection, analyses, or interpretation of data; in the writing of the manuscript, or in the decision to publish the results”.
References
Please reformat according template (and last published articles as example):
- Journal names abbreviated
- Exclude issue number
- Long dash for page range
- Year, volume and page range separated with commas
- Add doi
- Clear mistypes, e.g., Line 492
- Do not use pp. for journal articles
